# Object detection deep learning networks for Optical Character Recognition

## Abstract

In this article, we show how we applied a simple approach coming from deep learning networks for object detection to the task of optical character recognition in order to build image features taylored for documents. In contrast to scene text reading in natural images using networks pretrained on ImageNet, our document reading is performed with small networks inspired by MNIST digit recognition challenge, at a small computational budget and a small stride. The object detection modern frameworks allow a direct end-to-end training, with no other algorithm than the deep learning and the non-max-suppression algorithm to filter the duplicate predictions. The trained weights can be used for higher level models, such as, for example, document classification, or document segmentation.

## 1 Introduction

Document images make the use of deep learning networks a complex task, since most deep learning network architectures have been designed and trained for natural images, making them useless for document images which are mainly white and black characters and figures. This is in particular the case for classification networks (VGG, ResNets, ...), object detection networks (Fast RCNN, Faster RCNN, Yolo, SSD, ...), segmentation networks (FCN, U-Net, SegNet, DeconvNet, Dilated-Net, ParseNet, DeepLab...) which cannot be applied directly, even with finetuning.

Two challenges arise with deep learning and document data. First, we need to train specific features for the type of data. Second, the available datasets can be of smaller size than classical datasets for computer vision (ImageNet, COCO, ...), in particular when it is required to annotate images for a specific purpose.

To reduce the amount of data to train high level descriptions of the document images, such as document zones, segments, lines, or elements, the idea is to train a smaller network on OCR data which exists at massive scale, and use the weights of this small network as pretrained early layers in bigger networks to perform high level tasks with less required data.

In our experiments, we show that best performing approaches currently available for object detection on natural images can be used with success at OCR tasks. Code will be released on Github, so that the open research community can bring the best model architectures in terms of accuracy and speed/size efficiency.

## 2 Related work

In this section we quickly review the literature on OCR and object detection.

### 2.1 Approaches for OCR

Most deep learning approaches using Object Detection methods for OCR are applied to the task of scene text recognition also called text spotting, which consists in recognizing image areas of text, such as a sign or a wall plaque. Once the text area is recognized, a reading method is applied inside the zone. Some approaches use weakly supervised training either using a CTC loss leaving the alignment between the character positions and the output result to a recurrent network such as bidirectionnal LSTM (He et al. (2015), Jaderberg et al. (2014b), Wang et al. (2018), Goodfellow et al.

(2013), dro (2017), Liao et al. (2016)) or using a fixed number of softmax classifiers (Jaderberg et al. (2015), Bartz et al. (2017)) ; some other approaches use guided learning (He et al., 2018). These approaches are mainly driven by the Street View SVHN, Uber-Text (Zhang et al., 2017), FSNS (Smith et al., 2017), Coco-text (Veit et al., 2016), ICDAR 2003 (Lucas et al., 2003) and 2015 (Karatzas et al., 2015), SVT and IIIT5K (Mishra et al., 2012), Synth90k (Jaderberg et al., 2014a) datasets.

Rather than recognizing at word level or scene text level, few approaches concern direct detection of characters in natural images, using a localization network in ST-CNN (Jaderberg et al., 2015), or modern object detection approach in yolo-digits (Redmon & Farhadi, 2018) to recognize digits in natural images.

This work is the first to apply modern object detection deep learning approaches to document data with small convolutional networks, without converting them to natural images as in (Gilani et al., 2017). (Tensmeyer & Martinez, 2017) shows that document classification accuracy decreases with deeper networks.

## 2.2 Approaches for object detection

Modern object detections approaches are divided into two classes.

The first class yields to the highest accuracy object detectors, such as Fast-RCNN (Girshick, 2015), Faster-RCNN (Ren et al., 2015), Mask-RCNN (Detectron) (He et al., 2017), and is based on the two-stage approach of R-CNN (Girshick et al., 2013). In the first stage, an algorithm, such as Selective Search, or a deep learning model, generates a set of candidate proposals for object regions. In the second stage, a deep learning network classifies the presence of the object (the objectness), its class, as well as estimates the precise object bounding box.

In the second class of object detectors, the objectness, the class as well as the bounding box regression, are directly predicted by a single dense deep learning network. These approaches include OverFeat (Rowley et al., 1995), Yolo (Redmon et al. (2015), Redmon & Farhadi (2016), Redmon & Farhadi (2018)) or SSD (Liu et al., 2015).

## 3 Our approach to OCR

In our work, as a first attempt to use object detection networks to OCR, we design a single stage object detector, predicting the confidence of an object presence, the class, and the regression for the bounding box. In order to cope with multiple scales we use the feature pyramid approach of SSD (Liu et al., 2015).

### 3.1 Architectures

Our 1-scale models are inspired by the LeCun model for digit classification except that the dense layer have been converted to convolutions (locally linear) in order to compute a prediction at multiple positions in the image on a grid defined by the stride of the whole network. These models are composed of 2 convolution layers of kernel 3 and 32 and 64 features respectively, followed by a max pooling of stride 2 and 1 convolution layers of kernel 12 and 128 features, so that the receptive field of the network is 28 pixel large and wide. Offset of the model with valid paddings will be 14. We consider the stride of the last convolution as a parameter, $stride\_scale$, to adjust the stride of the whole model which will be $2 \times stride\_scale$. On top of these features, 4 stacks of dense layers are used for objectness, classification, position and scale regressions. We named this 1-scale model `CNN_C32_C64_M2_C128_D`.

Our 2-scale models are composed of 2 convolution layers of kernel 3 and 32 and 64 features respectively, followed by a max pooling of stride 2 and 2 other convolution layers of kernel 3 and 64 features and another max pooling of stride 2. Each max pooling layer is followed by a convolution layer of kernel 11 and 12 respectively, so that the receptive field for each output is 28 and 56 pixel large and wide. Offset for each output is 14 and 28 respectively. We consider the stride of the output convolutions as a variable parameter, $stride\_scale$, to adjust the stride of the whole model which will be $2 \times stride\_scale$ for the first output, and $4 \times stride\_scale$ for the second output. On top

of these 2 stages of features, 4 stacks of dense layers are used as well. We name this 2-scale model `CNN_C32_C64_M2_C64_C64_M2_C128_D_2`.

Each layer is followed by a ReLU activation, except for the outputs: objectness is computed with a sigmoid, classification with a softmax, position with hyperbolic tangent and scale with sigmoid.

1-scale model:

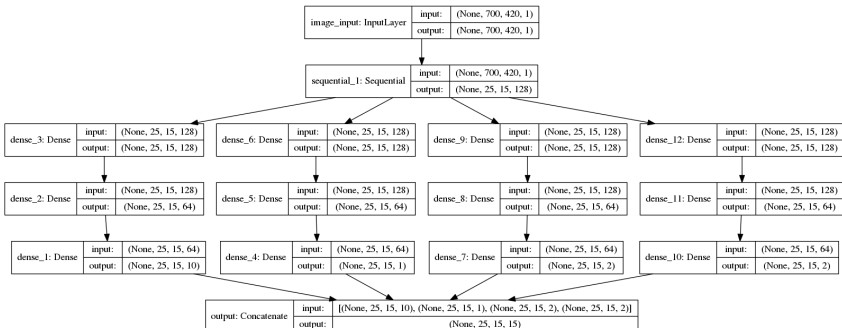

2-scale model:

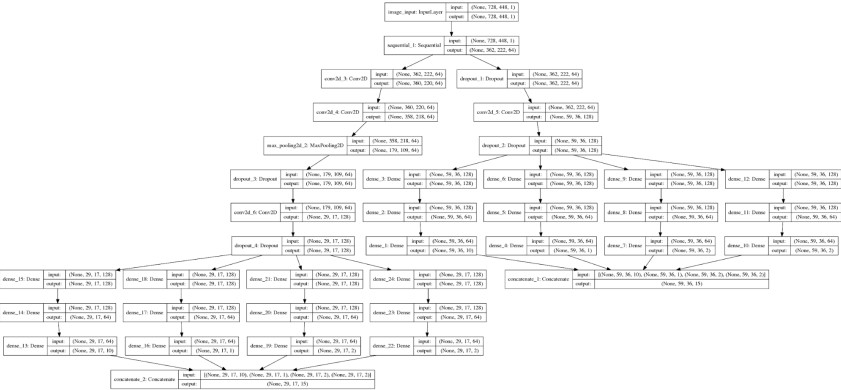

## 3.2 LOSS

For objectness, we need to consider the abundance of negative positions compared to positive positions. That is why we use the Tensorflow expression of weighted crossentropy designed to ensure stability and avoid overflow:

$$(1 - z) \times x + l \times \log(1 + \exp(- \mid x \mid)) + \max(-x, 0)$$

where $l = (1 + (q - 1) \times z)$ and $x = logits$, $z = targets$, $q = pos\_weight$. We found that a positive weight of 1000 works well on our OCR dataset.

The loss for classification and regression are crossentropy loss and MSE loss. We found that adding a multiplier of 100 to the regression loss help converge faster.

## 3.3 COMPUTATION OF AVERAGE PRECISION

It is common to use the mAP score as the final metric for object detection. In our case, we consider all classes as one class in order to use average precision as metric to measure the capacity of the models in terms of objectness and not classification. We use the name *object mAP* to distinguish it from classical mAP score.

The reason for this choice is that we focus on character detection in this work. For full character recognition, early results suggest that two-stage detectors might be of better fit than a one-stage detector, because in our 1-stage setting, classification accuracy drops when the classification network

is not well positioned on the character (see Stride experiments on Mnist), and this argument could give an advantage to 2-stage detectors.

Later on, we might add a second stage on top of this network as in Mask RCNN or Faster RCNN and this network might become a region proposal network. We leave this as future work, which purpose will be to improve the classification accuracy.

# 4 DATASETS

## 4.1 TOY DATASET

We build a toy dataset in order to test our implementations on a simpler task and check that the implementation is correct. For that purpose, we use the MNIST handwritten digits dataset to create pages with handwritten digits, at fixed or variable scales, with or without noise. The number of object classes is 10, the digits [”0”, ”1”, ”2”, ”3”, ”4”, ”5”, ”6”, ”7”, ”8”, ”9”].

Our MNIST dataset is composed of 1600 images of size 728x448, consisting of 28x28 square digits randomly placed on a 2D grid of stride 28.

| MNIST (default digit size 28) | MNIST with white prob .4 ( .9) | MNIST with noise | MNIST with digit size in 14-28 range |
|---|---|---|---|
| random digits at scale 28 at different positions on a grid | more digits per positions | cluttered with noise added randomly | random digit scale between 14 and 28. position is random |

| MNIST digit size 56 | MNIST with digit size in 28-56 range | MNIST with 2 digit sizes 28,58 | MNIST with 2 digit ranges 14-28,28-56 |
|---|---|---|---|
| random digits at scale 56 at different positions on a grid | random digits scale between 28 and 56. random position | digits at scales 28 and 56. Positions on a grid | random digit scales between 14 and 56. random positions. |

Our MNIST dataset with noise adds random distortions to create a high level of noise on the images and test the robustness of the models.

To check the performance of the position prediction, we set a different network stride, for example 12 (setting stride scale to 6), so that the network grid of positions where the model is evaluated in the convolutions, do not fall exactly on the grid of characters. That way, some digits will appear cropped, up to 6 pixels off horizontally and vertically, in the viewpoint of the network, ie its 28x28 receptive field.

To check the performance of the scale prediction, we build a MNIST dataset with digits randomly resized in the [14-28] range.

Before adding a layer to our network architecture as in SSD (Liu et al., 2015), we also check larger models at a bigger scale, with a MNIST dataset of 56 pixel wide digits.

Last, we build a two-scale dataset for two-layer predictions as in SSD (Liu et al., 2015), with digits at size 28 and 56, and add a middle output to `CNN_C32_C64_M2_C64_C64_M2_C128_D` to build `CNN_C32_C64_M2_C64_C64_M2_C128_D_2`, a two-scale network.

For a continuous scale between 14 pixels and 56 pixels, we build another two-scale dataset with 2 digit size ranges, 14-28 and 28-56.

## 4.2 OCR DATA

We build our own dataset of 8000 document pages, split into train (90%) and validation (10%) sets, for a fast check of our approach. Document PDF are converted to images with a resolution chosen automatically to have normal sized characters. To have fixed-sized image input for the network batched training, document images are then randomly cropped on a 728x448 area with characters, to have the same sized inputs as our mnist dataset.

We consider uppercase and lowercase letters, digits, the two parenthesis and the % sign. The number of classes is 65: [”0”, ”1”, ”2”, ”3”, ”4”, ”5”, ”6”, ”7”, ”8”, ”9”, ”a”, ”b”, ”c”, ”d”, ”e”, ”f”, ”g”, ”h”, ”i”, ”j”, ”k”, ”l”, ”m”, ”n”, ”o”, ”p”, ”q”, ”r”, ”s”, ”t”, ”u”, ”v”, ”w”, ”x”, ”y”, ”z”, ”A”, ”B”, ”C”, ”D”, ”E”, ”F”, ”G”, ”H”, ”I”, ”J”, ”K”, ”L”, ”M”, ”N”, ”O”, ”P”, ”Q”, ”R”, ”S”, ”T”, ”U”, ”V”, ”W”, ”X”, ”Y”, ”Z”, ”(”, ”)”, ”%”]

Letters are filtered by their size to fall in the range of [14-56] pixels and we start with two-scale networks ([14-28] and [28-56]) tested on our MNIST dataset.

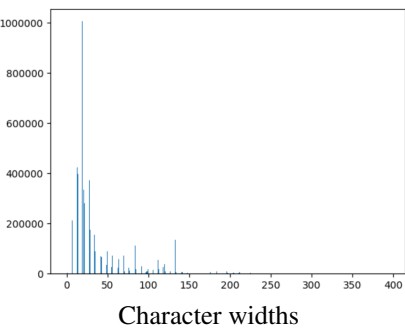 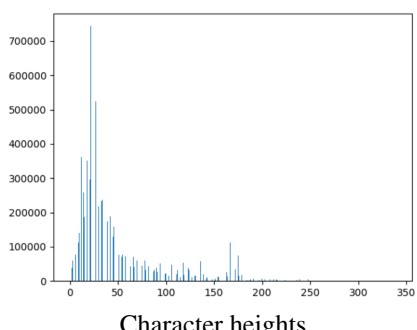

Character widths          Character heights

## 5 EXPERIMENTS

### 5.1 IMPLEMENTATION DETAILS

Code has been developed under Python with Keras deep learning framework, for Tensorflow and CNTK compute engines. It is compatible with Python 2.7 and 3.5 and allows multi-gpu training. For training, batch size is 3, the optimizer is Adam and the learning rate 0.001. Hyperparameters are searched by simple grid search. To create the OCR dataset, we use Tesseract OCR on 10 000 documents.

## 5.2 TOY DATASET

### 5.2.1 DIGITS CENTERED IN NETWORK FIELD OF VIEW

On the MNIST toy dataset, digits are always centered on a grid (of 28x28).

A 28-pixel-strided LeCun convolutional model offers a class accuracy above 99.2% since every positive position falls centered on the digit centered on a grid of stride 28. *object mAP* score is above 0.99 at 12 epochs with our simple model `CNN_C32_C64_M2_C128_D`.

With noise, *object mAP* score with our simple model `CNN_C32_C64_M2_C128_D` is above 0.98. Classification accuracy drops to 98.7.

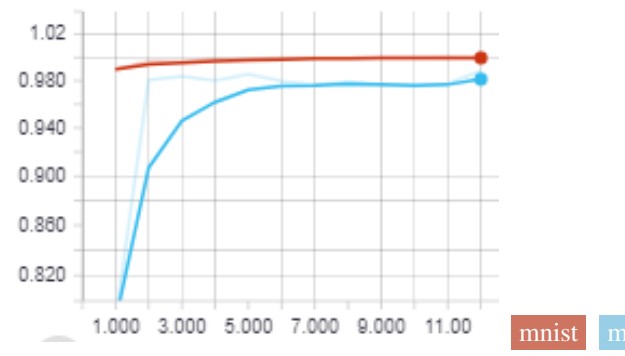

| Command | Obj acc | Class acc | Reg acc | Obj mAP |
|---|---|---|---|---|
| MNIST | 100 | 99.2827 | 1.60e-10 | 99.93 |
| MNIST with noise | 99.62 | 98.92 | 4.65e-6 | 98.41 |

### 5.2.2 THE EFFECT OF STRIDE AND IOU ON AVERAGE PRECISION

To test the network capability to predict position, we need to use a network stride different than the data grid stride. For example, with stride 12 instead of 28, most digits are not anymore in the center of the network reception field (except first row and column of our image).

Also, most digits will appear cropped in the field of view of the network and the IOU threshold defines how much crop ratio will be allowed to still consider the position on the grid as positive.

In the worst case, the network stride can be so large that some digits do not appear on the output grid, equivalent to an Intersection Over Union (IOU) of zero (intersection area is zero). Usually, the stride is not larger than the receptive field, and under this condition, the maximal intersection area between any digit and any network field is 50% times 50% = 0.25, while the union is 1.75, leading to a minimal IOU of 0.14. In the case of a smaller stride, for example 12 as below, the IOU threshold can be set higher without losing any digit for reconstruction:

| IOU | Obj acc | Class acc | Reg acc | Obj mAP | Target mAP |
|---|---|---|---|---|---|
| .15 | 96.37 | 36.25 | 0.010 | **99.97** | 100 |
| .2 | 98.42 | 28.56 | 0.012 | 99.75 | 100 |
| .25 | 97.05 | 36.42 | 0.015 | 99.52 | 100 |
| .3 | 98.35 | 92.78 | 0.0013 | 99.88 | 100 |
| .35 | 98.99 | 83.72 | 0.0069 | 99.22 | 100 |
| .4 | 98.70 | 94.96 | 0.0066 | 98.37 | 100 |
| .5 | 96.71 | 95.46 | 0.0062 | 91.09 | 95.71 |
| .6 | 99.92 | 98.23 | 4.8e-05 | 51.80 | 54.32 |
| .8 | 99.90 | **97.90** | 7.67e-05 | 8.5 | 10.63 |
| .95 | **99.94** | 97.27 | 3.7-07 | 10.80 | 12.21 |
| .99 | 99.91 | 97.66 | 7.06e-07 | 9.3 | 11.71 |

The large drop in classification accuracy for a small stride suggests that classification would benefit from better localized digits in the receptive field, which would encourage the use of 2-stage detectors.

To reduce the impact of the stride, we set a stride margin (see Experiment section on OCR data) on the digit max size to consider at a layer scale so that there is always one position on the network grid for which the character is fully seen by the network.

Reconstruction of ground truth from target data at 100% is only possible until an IOU threshold of 0.4, after which the network stride should be decreased. With a smaller stride of 4, reconstruction at 100% is possible at most IOU range:

| IOU | Obj acc | Class acc | Reg acc | Obj mAP | Target mAP |
|-----|---------|-----------|---------|---------|------------|
| .2 | 98.51 | 72.71 | 0.034 | 99.99 | 100 |
| .25 | 98.63 | 78.53 | 0.018 | **100** | 100 |
| .3 | 97.88 | 94.54 | 0.0098 | 99.89 | 100 |
| .4 | 96.85 | 97.41 | 0.0098 | 99.93 | 100 |
| .5 | 94.14 | 98.81 | 0.0099 | 99.61 | 100 |
| .6 | 99.80 | 98.57 | 0.00031 | 99.93 | 100 |
| .7 | 99.64 | 98.21 | 0.0016 | 99.77 | 100 |
| .8 | 100 | 98.19 | 1.7e-8 | 82.24 | 100 |
| .8 -e 30 | 99.98 | 99.35 | 1.73e-9 | 91.05 | 100 |

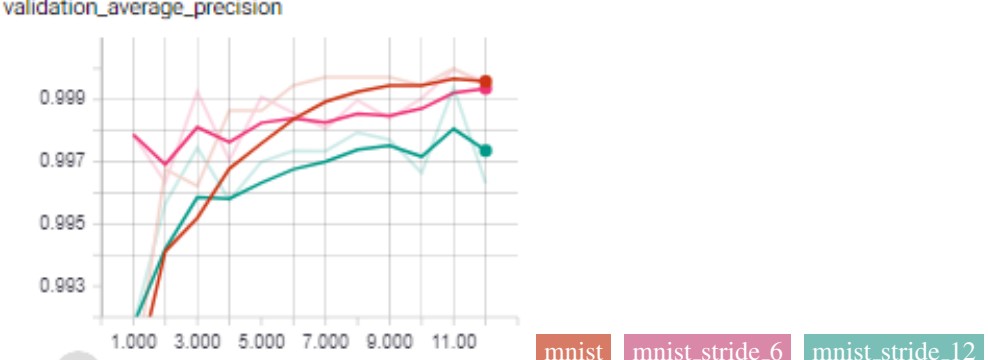

The images below show the target for an IOU of .2 for digits at scale between 7 and 14 pixels. The right image shows that with a large stride, small digits cut in the receptive field are dropped because of a too small IOU with the anchor, while with smaller stride, the IOU threshold does remove good candidates. A smaller stride enables to work with higher IOU and better mAP scores.

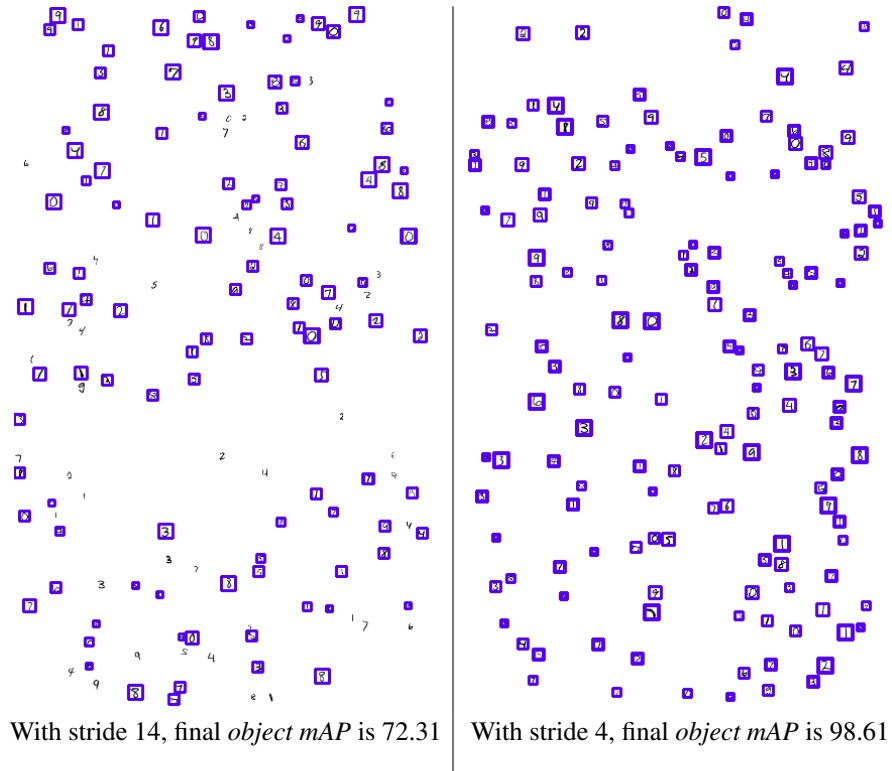

| With stride 14, final *object mAP* is 72.31 | With stride 4, final *object mAP* is 98.61 |

### 5.2.3 DIGIT SCALE PREDICTION

The second reason (after the cropped digits) to use a smaller IOU threshold is to capture small digits.

For example, for digits two times smaller, the maximal intersection of a digit with a network receptive field is 0.5 times 0.5 times 0.25 (the maximal intersection area for full size digits of 20), hence 0.0625, while the union is $1 + 0.5 \times 0.5 \times (1 - 0.25) = 1.1875$ leading to a minimal IOU of 0.052. About 3 times smaller than for the full digit size.

With a range scale of 14-28 for the digit sizes, the target *object mAP* (obtained when we reconstruct the bounding boxes from the target) remains 100% at IOU 0.25 for a stride of 12 pixels. The predicted *object mAP* is 99.58. The classification accuracy drops down to 89.37%.

### 5.2.4 HIGHER CAPACITY NETWORKS

Lets double the number of kernels to create `CNN_C64_C128_M2_C256_D` model. At stride 12 and IOU .3, classification accuracy increases from 92.78 to 96.84, while objectness remains roughly perfect. At stride 6 and IOU .2, it increases from 78.53 to 95.78%.

| Parameters | Obj acc | Class acc | Reg acc | Obj mAP | Target mAP |
|---|---|---|---|---|---|
| Stride 12, IOU .5 | 99.59 | 98.02 | 0.00078 | 92.32 | 94.89 |
| Stride 12, IOU .4 | 99.17 | 97.23 | 0.0047 | 99.79 | 100 |
| Stride 12, IOU .3 | 99.74 | 96.84 | 0.00043 | **100** | 100 |
| Stride 12, IOU .2 | 97.57 | 91.14 | 0.0016 | 99.98 | 100 |
| Stride 12, IOU .15 | 98.02 | 83.85 | 0.0083 | 99.95 | 100 |
| Stride 4, IOU .5 | 99.80 | 98.87 | 0.00053 | **100** | 100 |
| Stride 4, IOU .25 | 99.48 | 95.78 | 0.00054 | 100 | 100 |
| 14-28 pixel wide, Stride 12, IOU .25 | 96.58 | 91.42 | 0.0045 | 99.85 | 100 |

### 5.2.5 MULTI-STAGE NETWORKS

In order to capture digits in a bigger range than 28 pixels, we try networks with double reception field size, adding more layers (`CNN_C32_C64_M2_C64_C64_M2_C128_D` model), and possibly, multiple outputs at multiple layer stages (`CNN_C32_C64_M2_C64_C64_M2_C128_D_2` model) as in SSD (Liu et al., 2015).

First, we check our model with bigger field, `CNN_C32_C64_M2_C64_C64_M2_C128_D` model, on the MNIST dataset of 56 pixel wide digits. *object mAP* score is 1 while classification accuracy is 99.2% at 12 epochs, meaning this first architecture 56x56 receptive field deals well with digits twice big.

Then we add a second output to our network architecture as in SSD (Liu et al., 2015) to build `CNN_C32_C64_M2_C64_C64_M2_C128_D_2` model, and on a 2-scale dataset with digits at size 28 and 56, *object mAP* scores remain stable at 99.44 and 99.64 for network strides 12 and 4 respectively.

On a 2-scale dataset with digits at size ranges 14-28 and 28-56, *object mAP* score with our `CNN_C32_C64_M2_C64_C64_M2_C128_D_2` model is 98.82% and for the double size `CNN_C64_C128_M2_C128_C128_M2_C256_D_2` is 99.11%.

| Model | Digit size | Stride | IOU | Obj acc | Class acc | Reg acc | Obj mAP | Target mAP |
|---|---|---|---|---|---|---|---|---|
| S | 28-56 | 12 | .25 | 98.99 | 93.92 | 0.0018 | 99.89 | 100 |
| S | 14-28, 28-56 | 12 | .25 | 98.92 / 98.04 | 64.06 / 91.08 | 0.0037 / 0.0056 | 98.82 | 99.90 |
| D | 14-28, 28-56 | 12 | .2 | 98.57 / 97.73 | 58.30 / 79.84 | 0.0058 / 0.0036 | 98.31 | 99.90 |
| D | 14-28, 28-56 | 12 | .25 | 99.10 / 98.16 | 93.64 / 95.28 | 0.0016 / 0.0014 | 98.42 | 99.93 |
| D, 50e | 14-28, 28-56 | 12 | .25 | 99.26 / 98.78 | 93.91 / 94.02 | 0.0010 / 0.0014 | 98.81 | 99.93 |
| D, 50e | 14-28, 28-56 | 12 | .2 | 99.05 / 98.05 | 89.88 / 91.97 | 0.0021 / 0.0022 | 99.11 | 99.97 |
| S | 14-56 | 12 | .02 | 97.58 | 30.17 | 0.10 | 75.07 | 100 |
| S | 14-56 | 12 | .05 | 97.92 | 53.20 | 0.027 | 75.49 | 100 |
| S | 14-56 | 12 | .01 | 97.82 | 58.44 | 0.0057 | 87.45 | 92.67 |
| S | 14-56 | 12 | .2 | 98.82 | 79.23 | 0.0010 | 72.36 | 75.78 |

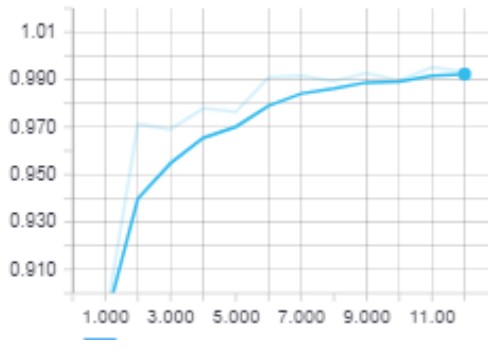

### 5.2.6 LOW RESOLUTION

In order to train full document image rather than a 700 pixel high crop of the images, resolution has to be smaller to fit in the GPU. For that reason, we look at models to recognize digits at a maximum size of 14 pixels instead of 28. We build a model `CNN_C32_C64_C128_D` by removing the max pooling layer. The network input fields becomes 14 pixel wide, and the stride is divided by 2.

With stride 8 after 30 epochs:

| Digit size | IOU | Obj acc | Class acc | Reg acc | Obj mAP | Target mAP |
|---|---|---|---|---|---|---|
| 14 | .3 | 97.12 | 94.50 | 0.012 | 99.91 | 100 |
| 7-14 | .2 | 98.58 | 73.07 | 0.0087 | 98.61 | 100 |
| 7-14 | .25 | 99.07 | 75.34 | 0.012 | 98.98 | 100 |

To capture digits on a larger range 7-28 pixel wide, we remove the 2 max pooling layers from our 56 pixel wide model, to build `CNN_C32_C64_C64_Cd64_C128_D`. At stride 3,

| Epochs | IOU | Obj acc | Class acc | Reg acc | Obj mAP | Target mAP |
|---|---|---|---|---|---|---|
| 30 | .1 | 97.47 | 73.70 | 0.010 | 87.19 | 95.45 |
| 30 | .2 | 99.08 | 92.84 | 0.0074 | 81.01 | 76.47 |
| 50 | .15 | 98.71 | 88.02 | 0.0046 | 87.79 | 84.76 |
| 50 | .1 | 97.97 | 79.19 | 0.0096 | 89.17 | 95.24 |

On 7-28 pixel digit range, the network sometimes learns a better reconstruction than the target, due to the hard IOU threshold decision in the target computation :

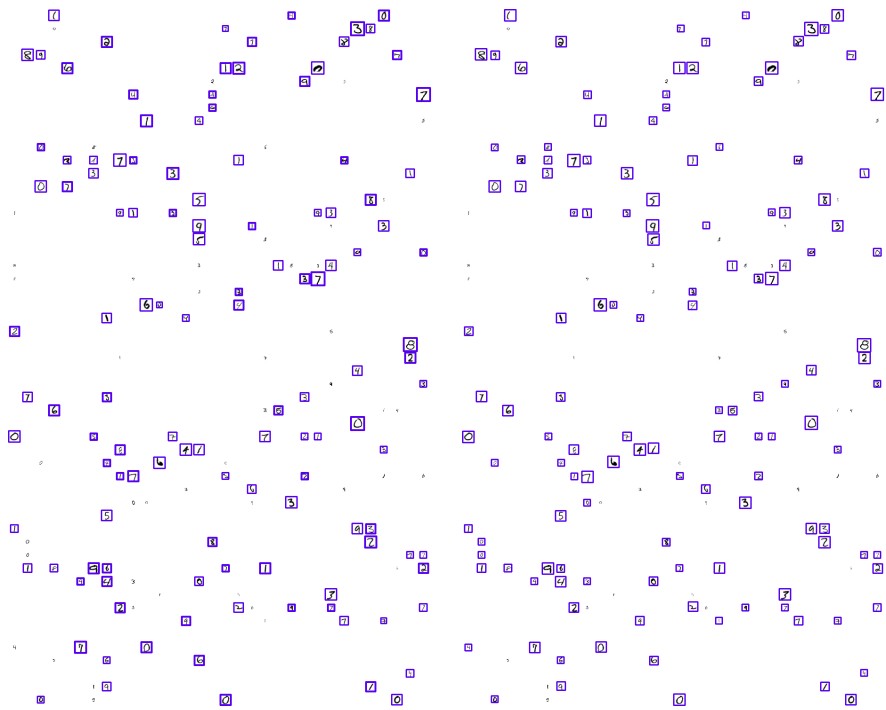

Targets mAP score is 76% at stride 8, IOU 0.2     While results mAP score is 80%

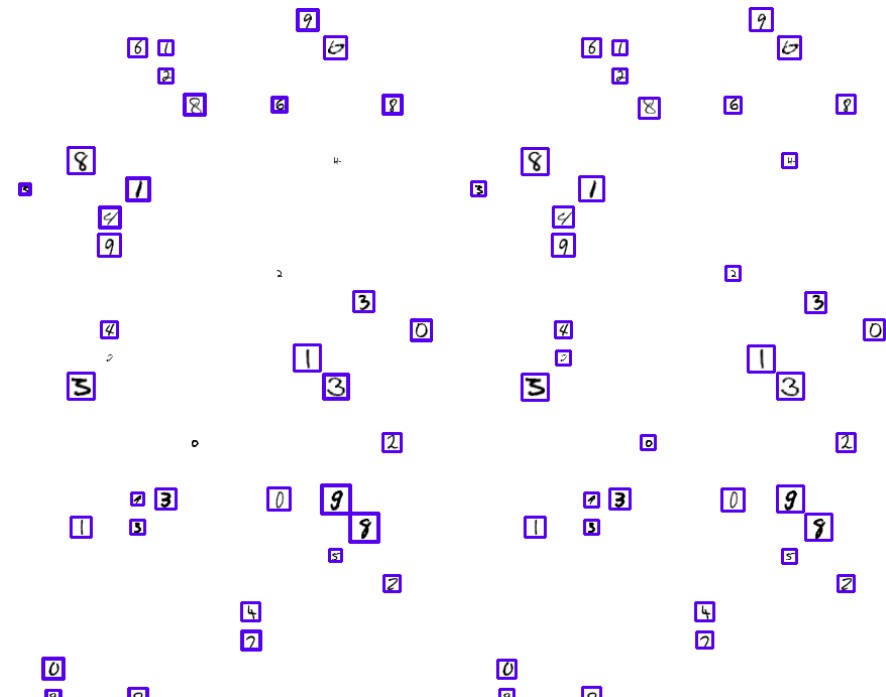

Targets mAP score is 86% at stride 8, IOU 0.15    While results mAP score is 89%

## 5.3 OCR DATASET

| Target (training data) | Detection results (with on the top left corner each layers receptive field minus stride margin) | Results filtered by NMS |
|---|---|---|
|  |  |  |

### 5.3.1 TARGET

We experiment different settings to define positives on the grid and compute the target average precision obtained if we reconstruct the bounding boxes from the target instead of the prediction results. We also compute the final average precision obtained by the trained model on this setting.

We consider positive a position on the grid that has a sufficient IOU with the receptive field of the network.

| Parameters | Obj acc | Class acc | Reg acc | Obj mAP | Target mAP |
|---|---|---|---|---|---|
| Stride 4+8, IOU 0.15 | 97.00 / 97.76 | 69.11 / 71.78 | 0.027 / 0.016 | 58.82 | 91.22 |
| Stride 4+8, IOU 0.2 | 97.89 / 98.44 | 75.39 / 72.75 | 0.020 / 0.011 | 68.09 | 84.47 |
| Stride 4+8, IOU 0.25 | 98.19 | 81.43 | 0.014 | **64.69** | 65.40 |
| Stride 6+12, IOU 0.15 | 97.52 / 97.58 | 72.18 / 77.03 | 0.028 / 0.015 | **67.05** | 86.07 |
| Stride 6+12, IOU 0.2 | 98.24 / 98.25 | 79.01 / 79.47 | 0.019 / 0.10 | 66.25 | 78.15 |
| Stride 6+12, IOU0.25 | 98.60 / 98.90 | 80.17 / 78.93 | 0.015 / 0.0075 | 62.71 | 66.42 |
| Stride 8+16, IOU 0.15 | 97.90 / 97.50 | 72.05 / 74.58 | 0.029 / 0.017 | 62.87 | 89.77 |
| Stride 8+16, IOU 0.2 | 98.42 / 97.99 | 78.35 / 79.15 | 0.021 / 0.012 | **66.30** | 83.94 |
| Stride 8+16, IOU 0.25 | 98.88 / 98.61 | 77.64 / 81.11 | 0.017 / 0.0077 | 60.26 | 69.35 |
| Stride 10+20, IOU 0.15 | 98.47 / 97.36 | 70.94 / 77.87 | 0.031 / 0.018 | **59.33** | 85.87 |
| Stride 10+20, IOU 0.2 | 98.92 / 97.76 | 67.94 / 80.13 | 0.021 / 0.014 | 51.87 | 77.52 |
| Stride 10+20, IOU 0.25 | 99.09 / 98.45 | 70.41 / 83.67 | 0.018 / 0.0097 | 44.59 | 61.57 |

IOU 0.2 (ce7562) IOU 0.15 (98cfe6) IOU 0.25 (de7ca2)

Target average precision is better when the IOU is low, since the grid misses no ground truth boxes, nevertheless, the model possibly learns better at an higher IOU, which also leads to better classification results.

We also tried considering as positive any character that fall in the receptive field of the network. Target average precision is very close to 1 but final average precision remains below 0.48.

### 5.3.2 STRIDE MARGIN

Since the network performs a strided analysis of the input image, we consider that characters should fall entirely into the receptive field of the network on one positive position. For that reason, we consider a stride margin, ie filter characters with a size lower than the receptive field dimension minus the stride. Deactivating this setting, some characters are not being seen completely by the network anymore and the prediction in position and scale should be harder to perform. *object mAP* score becomes 78.5%.

with stride margin (d0644c) without stride margin (6aa6c8)

### 5.3.3 POSITIVE WEIGHT AND LOSS WEIGHTS

pos weight 1000 (4ca89a) pos weight 100 (b9bfbf) pos weight (df9c83)

Best results are obtained with pos_weights=100.

### 5.3.4 KERNEL WIDTH

To see the influence of the width of the layers, we try to double the number of filters for each layer, leading to the `CNN_C64_C128_M2_C128_C128_M2_C256_D_2` model. A wider architecture does not seem to help much.

`CNN_C32_C64_M2_C64_C64_M2_C128_D_2` (f0f2f2) `CNN_C64_C128_M2_C128_C128_M2_C256_D_2` (b0d7d3)

Parameters Obj acc Class acc Reg acc Obj mAP Target mAP Stride 6+12, IOU 0.2 98.45/98.66 83.27/85.42 0.018/0.0097 70.11 78.15

### 5.3.5 FULL DOCUMENT, AT LOW RESOLUTION

Since document images are wide, we used a generator for the preprocessing, and do not have ground truth for mAP computation. Results are evaluated qualitatively. Best results with convolution of 28 pixel wide reception fields are obtained with images of max size 1000 pixels, since most characters fall at the right to be recognized.

At 1000 pixels:

While at size 1500 pixels, it misses the address :

More results at scale 1000: very small characters (below 7 pixels) as well as too big characters are missed, but main invoice information (amounts, headers, client,...) is recognized correctly.

## 6 CONCLUSION

Object detection architectures sound promising for high quality character reading and the development of document features through end-2-end training. Classical rule-based algorithms can reuse these features to solve higher level tasks, such as line extraction.

This opens the way for best model architectures search by the community. Future work includes reusing improvements from object detection models, such as multiple anchors, two-stage detection, focal loss, optimization tuning for larger images, batches or other input resolution.

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
