# OpenReview forum: "Object detection deep learning networks for Optical Character Recognition"
_ICLR.cc/2019/Conference_

### Official Review · AnonReviewer3 · 2018-11-02
**A definite incautious paper, no novelty and wrong formatting**

**Rating:** 2
**Confidence:** 5

**Review:**

This paper applied an object detection network, like SSD, for optical character detection and recognition. This paper doesn't give any new contributions and has no potential values.

weakness:
1. the paper is lack of novelty and the motivation is weak. I even can't find any contribution to OCR or object detection.

2. the paper is written badly so that I can't follow easily. In addition, the figures and tables are not always explained in the main body, which makes the experimental results confusing.

3. There are no titles in the figures and tables in this paper

4. the authors don't confirm the superiority of the proposed method to others.

minor comments
1. what's the meaning of Target mAP in the table?
2. It seems that Some figures are cropped from TensorBoard, with some extra shadows.

---

### Official Review · AnonReviewer2 · 2018-11-02
**A pure tech report with no novel contribution and serious flaws in the formatting, technical exposition and experimental evaluation**

**Rating:** 1
**Confidence:** 5

**Review:**

The authors experiment with building an SSD-like object detection network for text detection in documents, by replacing the usual VGG or ResNet base architecture with a light weight model inspired by the original digits classification CNN from [LeCun et al 1999].

This paper is a pure technical report with no novel contribution: all the authors do is replace the "body" network in the well-known SSD architecture with a simpler model (taken from existing literature) and evaluate it on two synthetic benchmarks of their creation.
The idea of employing object detection CNNs for OCR is not novel either, as pointed out in the related works section.

Beside the absence of novelty, the paper also suffers from several other serious flaws:

1) One of the main motivations provided by the authors for this work is that existing "classification [...] detection [...] or segmentation networks, cannot be applied directly, even with finetuning".
However, no experimental results are reported to justify this claim.
In fact, in the experimental section the proposed network is not compared against any existing baseline.

2) The text has serious clarity and formatting issues, in particular:
- most tables and figures have no caption, and the few that have one are not numbered
- the text exceed both the 8 pages limit and the extended 10 pages limit allowed in the case of big figures
- the experimental section is very confusing, in particular the way the authors refer to the various network variants using long code names makes it really hard for the reader to follow the ablation studies
- given the absence of proper captions and numbering, it is quite hard to understand which table refers to which experiment
- most of the graphs seem to be in the form of low-resolution bitmaps, which are quite hard to read even on screen
- many entries in the References section are either missing the venue, or point to an arXiv link even when a proper conference / journal reference would be available

3) Some important details about the network are missing, in particular the authors do not mention how labels are assigned to the network outputs, and only give a vague indication about the losses being used.
Similarly, there's no mention about the use of NMS, which is also an important component of the two architectures (SSD and YOLO) that inspire this work.
Assigning labels and performing NMS are actually some of the most crucial components in the training of object proposal / object detection networks, often requiring numerous meta-parameters to be properly configured and tuned, as testified by the meticulous descriptions given in previous works (e.g. YOLO and Fast / Faster / Mask r-CNN).

4) The experimental section is very poorly organized and formatted (as mentioned in (2) above), and completely lacks any comparison with other state of the art approaches.
A lot of space is devoted to presenting a detailed ablation study which, in my opinion, doesn't contribute much to the overall paper and actually reads more like a report on meta-parameter tuning.
Finally, starting from Section 5.3.1 the text seems to be copy-pasted without a second read from some differently formatted document, as entire phrases or possibly tables / figures seems to be missing.

In conclusion, in my opinion this paper does not meet the conference's minimum quality standards and should definitely be rejected.

---

### Official Review · AnonReviewer1 · 2018-11-06
**Object Detection for OCR**

**Rating:** 2
**Confidence:** 5

**Review:**

Unfortunately, the work does not introduce new contributions, with the point of the paper provided in the introduction:
In our experiments, we show that best performing approaches currently available for object detection
on natural images can be used with success at OCR tasks.

The work is applying established object detection algorithms to OCR. While the work provides a thorough experimental section exploring trade offs in network hyper-parameters, the application of object detection to the OCR domain does not provide enough novelty to warrant publication.

---

### Official Review · AnonReviewer4 · 2018-11-10
**A technical report about the application of well-known and standard deep learning techniques to character detection/recognition in document images**

**Rating:** 1
**Confidence:** 5

**Review:**

This paper lacks any novelty/contribution as it just applies well-known and standard architectures for object detection (SSD) and image classification (LeNet) trained with standard algorithms and losses.

Moreover, I fail to see what is the purpose of the proposed pipeline and it is not clear at all how it may help improving existing OCR engines in any particular scenario (handwriting recognition, printed text, historical documents, etc.). No demonstration or comparison with state of the art is provided.

The authors claim “This work is the first to apply modern object detection deep learning approaches to document data” but there are previously published works. For example:

Tuggener, Lukas, et al. "DeepScores--A Dataset for Segmentation, Detection and Classification of Tiny Objects." ICPR 2018.
Pacha, Alexander, et al. "Handwritten music object detection: Open issues and baseline results." DAS 2018.

Actually, in my opinion Music Object Detection in musical scores would be a much better test-bed/application for the proposed pipeline than any of the datasets used in this paper.  The datasets used in the experimental section seem to be created ad-hoc for the proposed pipeline and do not come from any real world application.

Finally, the presentation of the paper is marginal. Data plots have very bad resolution, there are no captions in any table or figure and they are not correctly referenced within the text. There seem to be also missing content in the last sections which makes them impossible to read/understand.

---

### Meta-Review · Area_Chair1 · 2018-12-13
**lacks novelty and clarity**

**Confidence:** 5
**Recommendation:** Reject

**Metareview:**

1. Describe the strengths of the paper.  As pointed out by the reviewers and based on your expert opinion.

The paper tackles an interesting and relevant problem for ICLR: optical character recognition in document images.

2. Describe the weaknesses of the paper. As pointed out by the reviewers and based on your expert opinion. Be sure to indicate which weaknesses are seen as salient for the decision (i.e., potential critical flaws), as opposed to weaknesses that the authors can likely fix in a revision.

- The authors propose to use small networks to localize text in document images, claiming that for document images smaller networks work better than standard SOTA networks for scene text. As pointed out in the reviews, the authors didn't make any comparisons to SOTA object detection networks (trained either on scene text or on document images) so their central claim has not been experimentally verified.
- The reviewers were unanimous that the work lacks novelty as object detection pipelines have already been used for OCR so a contribution of considering smaller detection networks is minor.
- There were serious issues with formatting and clarity.
These three issues all informed the final decision.

3. Discuss any major points of contention. As raised by the authors or reviewers in the discussion, and how these might have influenced the decision. If the authors provide a rebuttal to a potential reviewer concern, it’s a good idea to acknowledge this and note whether it influenced the final decision or not. This makes sure that author responses are addressed adequately.

There were no major points of contention and no author feedback.

4. If consensus was reached, say so. Otherwise, explain what the source of reviewer disagreement was and why the decision on the paper aligns with one set of reviewers or another.

The reviewers reached a consensus that the paper should be rejected.